# Effective Regularization with Relative-Distance Variances in Deep Metric Learning

## Abstract

This paper develops, for the first time, a novel method using relative-distance *variance* to regularize deep metric learning (DML), overcoming the drawbacks of existing pair-distance-based metrics, notably loss functions. Being a fundamental field in machine learning research, DML has been widely studied with the goal of learning a feature space where dissimilar data samples are further apart than similar ones. A typical approach of DML is to optimize the feature space by maximizing the relative distances between negative and positive pairs. Despite the rapid advancement, the pair-distance-based approach suffers from a few drawbacks that it heavily relies on the appropriate selection of margin to determine decision boundaries, and it depends on the effective selection of informative pairs, and resulting in low generalization across tasks. To address these issues, this paper explores the use of relative-distance variance and investigates its impact on DML through both empirical and theoretical studies. Based upon such investigation, we propose a novel Relative Distance Variance Constraint (RDVC) loss by regularizing the representation or embedding function learning. The proposed RDVC loss can seamlessly integrate with various pair-distance-based loss functions to ensure a robust and effective performance. Substantial experimental results have demonstrated the effectiveness of our proposed RDVC loss on both within-domain and cross-domain retrieval tasks. In particular, the RDVC loss is also shown useful in fine-grained zero-shot sketch-based image retrieval, a challenging task, revealing its general applicability to cross-domain and zero-shot learning.

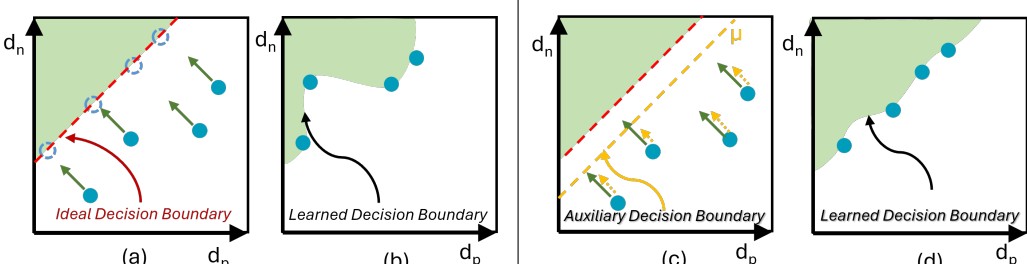

Figure 1: Concept illustration. *Distance-based Metrics (left)*: Optimizing positive and negative pairs to (a) an ideal decision boundary of $d_n - d_p = m$, but potentially result in (b) a distorted decision boundary. *Our idea (right)*: By regularizing the variance of relative distances, it provides (c) an auxiliary decision boundary of $d_n - d_p = \mu$ (where $\mu$ represents the mean relative distance for all pairs) to guide the optimization, achieving (d) the desired decision boundary.

## 1 Introduction

Deep Metric Learning (DML) represents one of the most influential fields in modern computer vision and machine learning research, receiving increasing attention as a result of the advancements in deep neural networks in recent years. DML has demonstrated remarkable performance in a variety of visual tasks, such as image recognition and retrieval. In the pursuit of learning a feature space that brings similar data samples closer while dissimilar ones further apart, researchers have introduced a range of metrics, i.e., loss functions. These losses typically utilize positive pair's distances $d_p$

and negative pair's distances $d_n$ to optimize the learned feature space by maximizing the relative distance $(d_n - d_p)$, where a fixed hyperparameter margin $m$ defines the desired gap between positive and negative samples. Examples of these loss functions include triplet loss (Hoffer & Ailon, 2015), contrastive loss (Hadsell et al., 2006), semi-hard triplet (SHT) loss (Schroff et al., 2015a), angular loss (Wang et al., 2017), hierarchical triplet loss (Ge, 2018), lifted structure loss (Oh Song et al., 2016), tuplet margin loss (Yu & Tao, 2019), ranked-list loss (Wang et al., 2019a) and multi-similarity (MS) loss (Wang et al., 2019b). By optimizing $(d_n - d_p)$, these methods aim to achieve a desired decision boundary $d_n - d_p = m$ (represented by the red dashed line in Fig. 1(a)). Nevertheless, because the actual relative distances between pairs may vary significantly due to different selected samples, it is challenging to reach the desired decision boundary (see Fig. 1(b)) by reducing relative distance by a fixed margin $m$ (represented by the green arrows of the same length in Fig. 1(a)). Some sampling methods were proposed to select more informative pairs (i.e. pairs are more representative of the embedding feature space) for training. For example, Schroff et al. (2015a) proposed a semi-hard mining method to select semi-hard samples while Ge (2018) built a hierarchical tree of all classes to collect hard negative pairs. Some other methods such as lifted structure loss, N-pair loss and MS-loss focus on assigning different weights for pairs based on their information. Nevertheless, selecting informative pairs is challenging, especially with large datasets, and large variation in relative distances can further complicate the optimization process.

In this paper, we investigate the use of relative-distance variances to address the above issues. To this end, we first examine the relationship between DML performance and the learned distribution of relative distances across different methods through an empirical study. Building upon the empirical study, we then propose a novel Relative Distance Variance Constraint (RDVC) loss to regularize the variance of relative distances across all pairs in the dataset. By adding an additional constraint to the optimization process, we introduce an auxiliary decision boundary of $d_n - d_p = \mu$ (indicated by the orange dashed line in Fig. 1(c)), where $\mu$ represents the mean relative distance of all pairs. Consequently, the RDVC loss facilitates the optimization of pairs toward the desired decision boundary, as illustrated in Fig. 1 (d). Moreover, the auxiliary decision boundary introduced by the RDVC loss at $d_n - d_p = \mu$ can reduce the reliance on selected margin $m$. If the chosen margin is far from the optimal value, the RDVC loss can guide the optimization toward $d_n - d_p = m$, enabling the identification of a desired decision boundary between $d_n - d_p = m$ and $d_n - d_p = \mu$. Furthermore, the RDVC loss makes the relative distance distribution more uniform by intentionally minimizing the variance of resulting distribution, increasing the possibility for selecting informative paris in the learning process. Hence, our main contribution is three-fold:

- We present a first-of-its-kind investigation, both empirical and theoretical, on the relative-distance *variance* and its impact on DML.
- Based upon such investigation, we introduce a novel loss function, $\mathcal{L}^{RDVC}$, which can seamlessly integrate with various pair-distance based loss functions to ensure robust and effective representation learning.
- Substantial experiments have been conducted to demonstrate the effectiveness of our proposed $\mathcal{L}^{RDVC}$ on both within-domain and cross-domain tasks, using three datasets: CUB200-2011, Cars196, and Sketchy. Particularly, our proposed $\mathcal{L}^{RDVC}$ exhibits a generic nature, enabling its application in the realm of fine-grained sketch-based image retrieval (FG-ZS-SBIR), which still represents a major challenge.

## 2 RELATED WORK

### 2.1 DISTANCE OPTIMIZATION AND REGULARIZATION

Being a rapid growing field of study in image retrieval research, deep metric learning (DML) focuses on learning an embedding space where similar samples are put closer together, while dissimilar samples are widely spaced. Research in this field can be broadly categorized into two groups, namely distance optimization and regularization.

**Distance optimization loss functions** aim to directly refine the relationships between sample pairs, triplets, or higher-order tuples, within the embedding space. The main objective is to minimize the distance between similar samples while maximizing the distance between dissimilar ones, typically using Euclidean, Mahalanobis or angular distances. For instance, the triplet loss (Schroff et al.,

2015a; Wu et al., 2017; Hoffer & Ailon, 2015) leverages anchor, positive, and negative samples to learn discriminative embedding space, whereby the anchor is closer to the positive sample than to the negative one. Different from triplet loss, contrastive loss (Hadsell et al., 2006; Chopra et al., 2005) operates on sample pairs, minimizing the distance between similar pairs, while penalizing dissimilar pairs within a specified range. To capture more complex relationships, higher-order variants (Chen et al., 2017) of these loss functions, such as binomial deviance loss (Yi et al., 2014), lifted structure loss (Oh Song et al., 2016), and multi-similarity loss (Wang et al., 2019b), have been developed. However, these methods, mainly optimizing pair-based distances, often suffer from slow convergence due to the quadratic increase in the number of sample pairs. To address this issue, proxy-based methods (Movshovitz-Attias et al., 2017; Aziere & Todorovic, 2019; Teh et al., 2020; Kim et al., 2020) have been introduced. They utilize class-level labels and learnable proxies as class centroids so as to streamline the optimization process and reduce computational complexity. Building on these pair-wise and class-level approaches, circle loss (Sun et al., 2020) combines the optimization of positive and negative pairs through dynamic weighting based on similarity, offering a unified paradigm. Nevertheless, these methods primarily rely on sample-level relative distances for informative sample utilization, requiring the careful selection of margin. Instead, we develop a simple, yet effective, loss losses to mitigate this problem and improve the performance of metric learning when combined with other existing losses.

**Regularization loss functions** aim to enhance the model's generalization capability by incorporating additional constraints during the learning process. While they might not directly optimize the distances between samples, they influence the learning process through regularization terms which promote a structured embedding space within this context. Zhang et al. (2020) proposed the spherical embedding constraint (SEC), which adaptively adjusts the embedding norms to lie on the same hypersphere, achieving more balanced directional updates and improved optimization stability. Roth et al. (2019), on the other hand, introduced mining interclass characteristics (MIC) to focus on interclass attributes, encouraging the model to learn more robust and discriminative feature representations. Roth et al. (2022) proposed non-isotropy regularization to enhance the robustness and generalization capability of the learned embeddings. Nevertheless, none of the above consider constraining the feature distribution from the relative distance point of view, which can eliminate the interference from the relative distance variability. To address the aforementioned issue and by analysing the gradient of loss function, we develop a novel RDVC loss function to regularize the relative distance.

## 2.2 WITHIN-DOMAIN AND CROSS-DOMAIN IMAGE RETRIEVAL

**Fine-grained image retrieval** The deep metric loss functions mentioned above are widely used in fine-grained image retrieval Schroff et al. (2015a); D'Innocente et al. (2021); Zhao et al. (2022). Compared to within-domain image retrieval, cross-domain image retrieval task is more challenging because input images are from different domains, such as photos and sketches. When applying the above loss functions to cross-domain retrieval task, particularly in zero-shot settings, these loss functions struggle to learn generalized features that effectively bridge the gap between different domains as well as between seen and unseen data.

**Fine-grained zero-shot sketch based image retrieval (FG-ZS-SBIR)**, involving zero-shot learning, deep metric learning, fine-grained retrieval as well as cross-domain adaption, is an extremely challenging task. Most of the existing methods developed from one of the above areas. Yu et al. (2016) first addressed this problem of fine-grained instance-level SBIR by constructing a shoe-and-chair dataset and using freehand sketches. In Sangkloy et al. (2016), a large dataset 'Sketchy' was provided as a benchmark for the research of FG-ZS-SBIR. To reduce the domain gap between sketch and photo, Shankar et al. (2018) leveraged multi-domain training data to train a classifier capable of generalizing across different domains while Pang et al. (2019) exploited an unsupervised learning approach to model a universal dictionary of prototypical sketches. Recently, with the development of large image models, such as DINO (Caron et al., 2021), CLIP (Radford et al., 2021), researchers improved the feature generalization for FG-ZS-SBIR using these large models. For example, Sain et al. (2023b) adopted the availability of unlabeled photo data to train a FG-ZS-SBIR model by means of semi-supervised method. Moreover, Sain et al. (2023a) and Lyou et al. (2024) adopted CLIP, a language-image pre-trained model, to use text semantic space to guide the learning of a highly versatile embedding space for FG-ZS-SBIR. To learn modality-specific features to distinguish between a sketch and a photo, Sain et al. (2023a) exploited prompt learning approach while Lyou et al. (2024) incorporated the modality encoder. Our method proposed in this paper, however, requires no

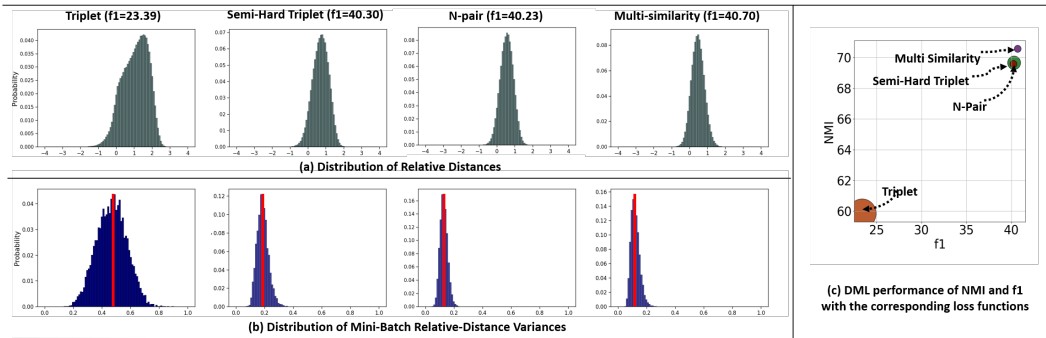

Figure 2: Empirical study results. (a) We train four baseline models using different loss functions and examine the resulting distribution of relative distances for the testing set of CUB200-2011 dataset (Wah et al., 2011). As shown, the model learned by the triplet loss shows a distinctly larger variance than the other 3, this also being reflected in the relevant DML performance measures (e.g., f1 score ↑). (b) We further estimate the mini-batch variance for different losses, the resulting expected values being indicated by red lines. This suggests that the DML performance can be improved by minimizing the mini-batch variance (e.g., below 0.2). (c) DML performance of different losses – symbol size represents the variance value.

additional training data nor a teacher model, in fact it does not even require changing the architecture of the model (i.e., parameter-free), thereby facilitating flexible integration with components from other studies.

## 3 METHOD

### 3.1 EMPIRICAL INVESTIGATION OF RELATIVE DISTANCE DISTRIBUTION.

We examine the relative-distance distributions learned by four models with triplet loss, semi-hard triplet (SHT) loss, pairwise loss, and MS-loss, respectively. These loss functions together represent nearly all existing strategies of improving DML performance with pair labels. Assume an anchor sample $x$ has $K$ positive samples $\{x_p^i\}_{i=1}^K$ of the same category and $L$ negative samples $\{x_n^j\}_{j=1}^L$ from other categories. The positive and negative distances are denoted as $\{d_p^i\}_{i=1}^K$ and $\{d_n^j\}_{j=1}^L$, respectively. We train four baseline models with the four losses, which their definitions are given below.

**Triplet loss** aims to force distance between the anchor sample $x$ and a negative sample $x_n^j$ to be larger than that of a randomly selected positive one $x_p^i$ over a given margin $m$:

$$\mathcal{L}_{tri} = [d_p^i - d_n^j + m]_+ \tag{1}$$

**Semi-Hard Triplet (SHT) loss** focuses on selecting triplets where the negative sample $x_n^j$ is far from the anchor than the positive sample $x_p^i$:

$$\mathcal{L}_{SHT} = [d_p^i - d_n^j + m]_+ \quad \forall \quad \{d_p^i, d_n^j\} \quad \text{s.t.} \quad d_p^i < d_n^j < d_p^i + m \tag{2}$$

**N-pair loss** utilizes multiple negative samples for each positive pair as follows:

$$\mathcal{L}_{NP} = -\log\left(\frac{e^{1-d_p^i}}{e^{1-d_p^i} + \sum_{j=1}^L e^{1-d_n^j}}\right) \tag{3}$$

**MS-loss** leverages not only multiple negative samples but also multiple positive samples for each anchor sample, and assign different weights to pairs based on the relative distances between positive and negative pairs. Additionally, a pair mining strategy is introduced to select informative

positive and negative pairs, drawing inspiration from the Large Margin Nearest Neighbor (LMNN) approach (Weinberger et al., 2005). MS-loss is defined as:

$$\mathcal{L}_{MS} = \frac{1}{\alpha} \log \left( 1 + \sum_{i=1}^{K} e^{-\alpha(1-d_p^i-m)} \right) + \frac{1}{\beta} \log \left( 1 + \sum_{j=1}^{L} e^{\beta(1-d_n^j-m)} \right) \quad (4)$$

$$\forall (d_p^i, d_n^j) \quad \text{s.t.} \quad 1 - d_n^i > min\{1 - d_p^i\}_{i=1}^{K} - \epsilon \quad \text{and} \quad 1 - d_p^j < max\{1 - d_n^j\}_{j=1}^{L} + \epsilon$$

where $\epsilon$ is a given hyperparameter for LMNN selection.

**Result Analysis.** The results of the empirical study are shown in Fig. 2, where the details of the study are provided in the figure caption. We calculate their distances with cosine similarity, and then the relative distances between positive and negative pairs for each anchor sample in the test set of CUB200-2011 dataset (Wah et al., 2011). The relative distance distribution of these losses are shown in Fig. 2(a). Since models are optimized per mini-batch, we also calculate the variance of mini-batch relative distances and visualize their distributions across mini-batches in Fig. 2(b). As shown, it is evident that the triplet loss exhibits a significantly larger variance compared to the others (i.e., semi-hard triplet (SHT), N-pair and MS losses). Surprisingly, although the motivation and internal design of SHT and N-pair losses are very different (Eqs. (2) and (3)), the distributions of their learned relative distance exhibit significant similarities, as shown in Fig. 2(b). Furthermore, these observed patterns are also present in response to the batch-wise estimator distribution. Last but not least, the variances exhibit a negative correlation with the performance measures of DML, namely the f1 and Normalized Mutual Information (NMI) scores in this case (Fig. 2(c)).

### 3.2 *RDVC* LOSS AND THEORETICAL GRADIENT ANALYSIS

Assume a mini-batch with $N$ pairs of training images $\{(x^1, x_p^1; y^1), (x^2, x_p^2; y^2), ..., (x^N, x_p^N; y^N)\}$, where each pair of anchor and positive samples $< x^i, x_p^i >$ associates with a distinct category $y^i$. Then, a triplet can be represented as $< x^i, x_p^i, x^{\neq i} >$. The relative distance for each triplet is obtained as follows:

$$\mathcal{D}_i = d_p^i - d_n^i = d(f_a^i, f_p^i) - d(f_a^i, f_n^i) \quad (5)$$

where $f_a^i, f_p^i$ are the features of anchor sample $x^i$ and positive sample $x_p^i$, respectively. $f_n^i$ is the feature of a negative sample that can be sampled from $x^{\neq i}$. $d(\cdot)$ is the distance function (e.g., L2 or cosine distance). We propose a new loss, named relative distance variance constraint (RDVC ) loss, to regularize the resulting relative-distance distribution by minimising its variance as follows

$$\mathcal{L}^{RDVC} = \hat{\sigma}^2(\mathcal{D}) = \frac{1}{N-1} \sum_{i=1}^{N} (\mathcal{D}_i - \hat{\mu}_{\mathcal{D}})^2 \quad \text{where} \quad \hat{\mu}_{\mathcal{D}} = \frac{1}{N} \sum_{i=1}^{N} \mathcal{D}_i \quad (6)$$

The total loss is obtained by combining the triplet loss with its associated RDVC loss as follows:

$$\mathcal{L}^{total} = \lambda_1 \mathcal{L}^{RDVC} + \mathcal{L}^{tri} \quad (7)$$

Hereafter, we conduct a theoretical gradient analysis by deriving RDVC (left component of Eq. (7)) w.r.t. $\mathcal{D}_i$ as follows:

$$\frac{\partial \mathcal{L}^{RDVC}}{\partial \mathcal{D}_i} = \frac{\partial}{\partial \mathcal{D}_i} \frac{1}{N-1} \sum_{j=1}^{N} (\mathcal{D}_j - \hat{\mu}_{\mathcal{D}})^2 = \frac{2}{N-1} (\mathcal{D}_i - \hat{\mu}_{\mathcal{D}}) \quad (8)$$

Since $\frac{\partial \mathcal{D}_j}{\partial \mathcal{D}_i} = 0$ for $i \neq j$, $\frac{\partial \mathcal{D}_j}{\partial \mathcal{D}_i} = 1$ for $i = j$, and $\frac{\partial \hat{\mu}_{\mathcal{D}}}{\partial \mathcal{D}_i} = \frac{1}{N}$.

The gradient of the triplet loss $\mathcal{L}^{tri}$ (right component of Eq. (7)) is as follows:

$$\frac{\partial \mathcal{L}^{tri}}{\partial \mathcal{D}_i} = \frac{\partial}{\partial \mathcal{D}_i} \max(\mathcal{D}_i + \alpha, 0) = \begin{cases} 1, & \text{if } \mathcal{D}_i \geq -\alpha \\ 0, & \text{otherwise} \end{cases} \quad (9)$$

From Eq. (9), it can be seen that for an easy triplet (i.e., when $\mathcal{D}_i < -\alpha$), the gradient is consistently 0. Otherwise, for a hard triplet (i.e. when $\mathcal{D}_i \geq -\alpha$), the gradient remains constant with a value of **1**, regardless the variance ($\sigma(\mathcal{D})$) within the mini-batch. This could lead to an error as a more difficult triplet should receive a higher gradient compared to a less difficult one. Such gradient discrepancies can hinder the learning process. Nevertheless, by taking into account Eq. (8), this error can be avoided, since it proposes that if a more difficult triplet deviates from the mean value within a mini-batch, the gradient of $\mathcal{L}^{RDVC}$ is added to facilitate the

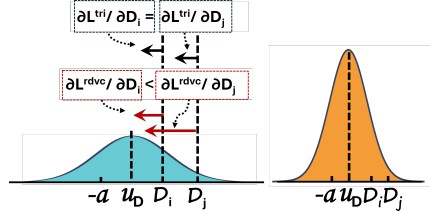

Figure 3: (a) relative-distance distribution of $\mathcal{L}^{tri}$, (b) resulting distribution with combined gradient effect of RDVC and triplet losses.

optimization of $\mathcal{L}^{tri}$. Fig. 3 illustrates our learning process, with the lengths of the **red** and **black** solid arrows indicating the norm of the gradients for $\mathcal{L}^{RDVC}$ and $\mathcal{L}^{tri}$, respectively, in (a). When considering the various distances represented by $D_i$ and $D_j$, the gradient magnitudes of $\mathcal{L}^{RDVC}$ differ from those of $\mathcal{L}^{tri}$. As a consequence, the resulting distribution becomes more uniform (Fig. 3(b)), leading to enhanced model generalization. Note that for the most difficult triplet, our RDVC loss will create the greatest acceleration $\frac{2}{N-1}(\max(\{\mathcal{D}_{i=1}^N\}) - \hat{\mu}_{\mathcal{D}})$. On the other hand, if the given triplet is less difficult ($\mathcal{D}_i$ vs $\mathcal{D}_j$), our $\mathcal{L}^{RDVC}$ also provides a guidance for this triplet, in order to push $\mathcal{D}_i$ more closer to the averaged relative distance $\hat{\mu}_{\mathcal{D}}$. This means that we are striving for a difficulty-uniform feature space by adjusting all samples, appropriately. (See supplementary materials for another illustrative example.)

### 3.3 INTEGRATING OTHER LOSS FUNCTIONS

For simplicity, we have discussed our proposed loss function when the given mini-batch consists of positive pairs under different categories. It is not difficult to see that, if the class distribution is balanced and the sample distribution is uniform, the mini-batch variance obtained by Eq. (6) serves as an unbiased estimator (Hermans et al., 2017), i.e., $\mathcal{L}^{RDVC} = \mathbb{E}(\hat{\sigma}^2) = \sigma^2$.

In addition to triplet loss, the proposed RDVC loss can integrate with other existing loss functions to ensure a robust and effective representation learning. In the experiment section, we will demonstrate the integration of RDVC with four well-known DML loss functions, including triplet (Hoffer & Ailon, 2015), semi-hard triplet (SHT) (Schroff et al., 2015a), N-pair (NP) (Sohn, 2016), and multi-similarity (MS) (Wang et al., 2019b) losses. Moreover, we utilize the SEC loss ($\mathcal{L}^{SEC}$) to better normalize each feature. Hence, our toal loss function is given by:

$$\mathcal{L}^{total} = \lambda_1 \, \mathcal{L}^{RDVC} + \eta \, \mathcal{L}^{SEC} + \begin{cases} \mathcal{L}^{tri}, & \text{if triplet loss is selected} \\ \mathcal{L}^{SHT}, & \text{if semi-hard triplet loss is selected} \\ \mathcal{L}^{NP}, & \text{if N-pair loss is selected} \\ \mathcal{L}^{MS}, & \text{if Multi-Similarity loss is selected} \end{cases} \quad (10)$$

where $\lambda_1$ is our hyperparameter and $\eta$ is the hyperparameter for SEC loss (Zhang et al., 2020). It is noteworthy that we offer the advantage complementing existing losses instead of replacing them entirely, which has proven advantages in various applications (Zhang et al., 2020). Our supplemental materials provide a full version of theoretical gradient analysis for other losses and the implementation details.

## 4 EXPERIMENTS

We comprehensively evaluated the effectiveness of our RDVC on both within-domain and cross-domain image retrieval tasks, using fine-grained image retrieval datasets for the former while applying our RDVC on the FG-ZS-SBIR dataset for the latter. The latter represents a challenging task due to the large visual gap between sketches and real photos and the requirement for fine-grained discrimination between visually similar categories.

## 4.1 DATASETS AND EXPERIMENTAL SETTINGS

**Fine-grained image retrieval.** We evaluated our method on two benchmark datasets: CUB200-2011 (Wah et al., 2011) and Cars196 (Krause et al., 2013). The CUB200-2011 dataset contains 11,818 photos covering 200 categories of birds, with the first 100 categories consisting of 5,894 photos being used for training purposes, and the other 100 categories, with 5,924 photos, for testing. The Cars196 dataset has 196 different categories, with 16,183 photos of cars in total. We used the first 98 categories for training purposes and the other 98 categories for testing. Our experimental work was based upon that of Zhang et al. (2020) in that we used the BN-Inception (Ioffe, 2015) as the backbone network and initiated the model weights from an ImageNet pre-trained model. The PK sampling strategy (Hermans et al., 2017) was adopted to construct the mini-batches and the $P$ and $K$ values for the different datasets are in accordance with those of Zhang et al. (2020). The batch size was set at 120 and embedding size at 512 for all the methods and datasets, while the Adam optimizer was used to optimize the loss function. For $\eta$ in $\mathcal{L}^{total}$, we adopted the values used in the other study (Zhang et al., 2020) for the various datasets and loss functions. Readers may refer to the supplemental materials for details.

**FG-ZS-SBIR.** We used the popular Sketchy dataset (Sangkloy et al., 2016) to evaluate our method for FG-ZS-SBIR. The Sketchy dataset contains 125 categories, each with 100 photos and at least 5 fine-grained sketches. To in line with Yelamarthi et al. (2018), we splitted the dataset into 104 categories for training purposes and 21 categories for testing. Various networks pre-trained on ImageNet, including InceptionV3, P-ViT and ViT, are used as our backbone feature extractors. The input size is set as $224 \times 224$ and the batch size as 64. The model is trained using the Adam optimizer with a learning rate of $lr = 1e-4$; with the other learning hyperparameters $\beta_1 = 0.9$ and $\beta_2 = 0.999$. To preserve the knowledge of pre-trained models, all the parameters of the models are frozen, except for the layer normalization during the training stage.

**Performance Measures.** For fine-grained image retrieval, we evaluate performance using Normalized Mutual Information (NMI), F1 score, and retrieval rates at R@1, R@2, R@4, and R@8. For FG-ZS-SBIR, we follow Sain et al. (2023a) and evaluate effectiveness using Acc@1, Acc@5, and Acc@10.

Table 1: The compatibility of the RDVC loss with other losses.

| Loss | CUB200-2011 Dataset | | | | | | Cars196 Dataset | | | | | |
|---|---|---|---|---|---|---|---|---|---|---|---|---|
| | NMI | F1 | R@1 | R@2 | R@4 | R@8 | NMI | F1 | R@1 | R@2 | R@4 | R@8 |
| Triplet | 59.85 | 23.39 | 53.34 | 65.60 | 76.30 | 84.98 | 56.66 | 24.44 | 60.79 | 71.30 | 79.47 | 86.27 |
| Ours: Triplet+RDVC | **67.01** | **34.97** | **58.32** | **70.88** | **81.74** | **88.83** | **64.74** | **33.70** | **76.37** | **84.65** | **90.33** | **94.29** |
| Triplet+SEC | 64.24 | 30.83 | 60.82 | 71.61 | 81.40 | 88.86 | 59.17 | 25.51 | 67.89 | 78.56 | 85.59 | 90.99 |
| Oours: Triplet+SEC+RDVC | **68.10** | **37.62** | **62.31** | **74.27** | **83.88** | **90.61** | **67.89** | **38.78** | **79.89** | **88.07** | **93.09** | **96.04** |
| SHT | 69.66 | 40.30 | 65.31 | 76.45 | 84.71 | 90.99 | 67.64 | 38.31 | 80.17 | 87.95 | 92.49 | 95.67 |
| Ours:SHT+RDVC | **71.34** | **42.81** | **66.91** | **77.38** | **85.60** | **91.81** | **71.46** | **44.47** | **83.63** | **90.33** | **94.17** | **96.59** |
| SHT+SEC | 71.62 | 42.05 | 67.35 | 78.73 | 86.63 | 91.90 | 72.67 | 44.67 | 85.19 | **91.53** | **95.28** | **97.29** |
| Ours: SHT+SEC+RDVC | **73.53** | **46.30** | **68.15** | **78.88** | **87.15** | **92.34** | **73.31** | **46.05** | **85.50** | 91.35 | 95.07 | 97.10 |
| N-pair | 69.58 | 40.23 | 61.36 | 74.36 | 83.81 | 89.94 | 68.07 | 37.83 | 78.59 | 87.22 | 92.88 | 95.94 |
| Ours: N-pair+RDVC | **71.01** | **41.90** | **64.94** | **76.01** | **84.49** | **91.14** | **71.02** | **42.32** | **82.57** | **89.40** | **94.38** | **97.01** |
| N-pair+SEC | 72.24 | 43.21 | 66.00 | 77.23 | 86.01 | 91.83 | 70.61 | 42.12 | 82.29 | 89.60 | 94.26 | 97.07 |
| Ours: N-pair+SEC+RDVC | **73.30** | **45.62** | **67.69** | **79.42** | **87.39** | **92.47** | **72.43** | **44.30** | **83.59** | **90.25** | **94.79** | **97.37** |
| MS | 70.57 | 40.70 | 66.14 | 77.03 | 85.43 | 91.26 | 70.23 | 42.13 | 84.07 | 90.23 | 94.12 | 96.53 |
| Ours: MS+RDVC | **71.05** | **42.57** | **66.59** | **77.60** | **85.47** | **91.48** | **71.98** | **44.74** | **84.52** | **90.54** | **94.40** | **96.83** |
| MS+SEC | 72.13 | 42.60 | 68.77 | 79.37 | 87.15 | 92.08 | 73.04 | **47.17** | 84.93 | 91.28 | 95.03 | 97.17 |
| Ours: MS+SEC+RDVC | **73.94** | **46.73** | **69.48** | **80.11** | **87.49** | **92.47** | **73.13** | 45.55 | **86.95** | **92.58** | **95.82** | **97.77** |

## 4.2 FINE-GRAINED IMAGE RETRIEVAL

We consider five representative baseline loss functions, namely, triplet loss, semihard triplet loss (SHT), N-pair loss, spherical embedding constraint loss (SEC), multi-similarity loss (MS), as well as a norm feature regularization loss function to evaluate our RDVC . In Table 1, we compare these loss functions, with and without RDVC , on the CUB200-2011 (Wah et al., 2011) and Cars196 (Krause et al., 2013) datasets. Table 1 shows that RDVC improves the performance of these baseline loss functions on both datasets, indicating that reducing the variance of relative distance is effective in improving performance. For example, compared to the triplet loss, the use of RDVC significantly

Table 2: Comparison with SOTA methods (with BN-Inception).

| Methods | CUB200-2011 dataset | | | | Cars196 dataset | | | |
|---|---|---|---|---|---|---|---|---|
| | R@1 | R@2 | R@4 | NMI | R@1 | R@2 | R@4 | NMI |
| HTL (Ge, 2018) | 57.1 | 68.8 | 78.7 | - | 81.4 | 88.0 | 92.7 | - |
| RLL-H (Wang et al., 2019a) | 57.4 | 69.7 | 79.2 | 63.6 | 74.0 | 83.6 | 90.1 | 65.4 |
| MS (Wang et al., 2019b) | 65.7 | 77.0 | 86.3 | - | 84.1 | 90.4 | 94.0 | - |
| SoftTriple (Qian et al., 2019) | 65.4 | 76.4 | 84.5 | 69.3 | 84.5 | 90.7 | 94.5 | 70.1 |
| GroupLoss (Elezi et al., 2020) | 65.5 | 77.0 | 85.0 | 69.0 | 85.6 | 91.2 | 94.9 | |
| CircleLoss (Sun et al., 2020) | 66.7 | 77.4 | 86.2 | - | 83.4 | 89.8 | 94.1 | - |
| ProxyAnchor (Kim et al., 2020) | 68.4 | 79.2 | 86.8 | - | 86.1 | 91.7 | 95.0 | - |
| ProxyGML (Zhu et al., 2020) | 66.6 | 77.6 | 86.4 | 69.8 | 85.5 | 91.8 | 95.3 | 72.4 |
| DRML (Zheng et al., 2021) | 68.7 | 78.6 | 86.3 | 69.3 | 86.9 | 92.1 | 95.2 | 72.1 |
| HIST (Lim et al., 2022) | **69.7** | 80.0 | 87.3 | 70.8 | **87.4** | 92.5 | 95.4 | 73.0 |
| Ours: MS+SEC+RDVC | 69.5 | **80.1** | **87.5** | **73.0** | 87.0 | **92.6** | **95.8** | **73.1** |

Table 3: Performance comparison for the FG-ZS-SBIR task on the Sketchy.

| Methods | Backbone | Acc@1 | Acc@5 | Acc@10 |
|---|---|---|---|---|
| Hard-Transfer (Yu et al., 2016) | | 16.0% | 40.5% | 55.2% |
| CVAE-Regress (Yelamarthi et al., 2018) | | 2.4% | 9.5% | 17.7% |
| Reptile (Nichol & Schulman, 2018) | IN-V3 | 17.5% | 42.3% | 57.4% |
| CrossGrad (Shankar et al., 2018) | | 13.4% | 34.9% | 49.4% |
| CC-DG (Pang et al., 2019) | | 22.7% | 42.1% | 63.3% |
| Ours: Triplet+RDVC | | **26.3%** | **53.4%** | **66.5%** |
| SketchPVT (Sain et al., 2023b) | P-ViT | 30.2% | 51.7% | - |
| Ours: Triplet+RDVC | | **32.9%** | **60.4%** | 72.4% |
| CLIP-AT (Sain et al., 2023a) | | 28.7% | **62.3%** | - |
| MARL (Lyou et al., 2024) | ViT | 29.8% | 57.9% | - |
| Ours: Triplet+RDVC | | **31.0%** | 60.4% | 73.1% |

increases the NMI, F1 and R@1 by 7.16%, 11.58% and 4.98%, respectively, on CUB200-2011. Compared to different losses in Table 1, RDVC has shown improved performance on the NMI, F1 and R@1 of the most state-of-the-art model with MS-loss by 1.75%, 2.09% and 0.45%, respectively. Furthermore, combining SEC and RDVC can further improve performance. Specifically, on the CUB200-2011 dataset, using RDVC increases the NMI, F1 and R@1 wehn integrated with both triplet and SEC losses by 3.86%, 6.79% and 1.49%, respectively. On the Cars196 dataset, with the combination of MS and SEC, RDVC increases the NMI, F1 and R@1 by 0.96%, 2.23% and 0.62%, respectively. This demonstrates that RDVC, which focuses on reducing the variance of relative distance, effectively complements SEC, which focuses on reducing the variance of feature norms. In Table 2, we also compare our method with other 10 SOTA methods in metric learning on both the CUB200-2011 and Cars196 datasets. It can be seen that our method always achieves nearly the highest accuracy in all metrics on both datasets, indicating its outstanding performance in detecting differences between fine-grained image categories as well as its robustness to noisy or widely varied samples. The R@1 and R@2 of our method are only slightly lower than those of HIST. Nevertheless, unlike HIST, which relies on graph networks to utilize multilateral semantic relations, our method does not require the addition of new networks for training.

## 4.3 FG-ZS-SBIR

In Table 3, we evaluate our RDVC loss for fine-grained sketch-based image retrieval (FG-ZS-SBIR) in terms of the triplet loss with our RDVC loss, compared to eight SOTA methods. Table 3 shows that our model outperforms all the listed methods. In particular, our method outperforms the best method of CC-DG (Pang et al., 2019), among the InceptionV3-based models, by 3.7% and 11.3% in terms of Acc@1 and Acc@5, respectively. Furthermore, our method outperforms SketchPVT, which utilizes additional photos for training, by 2.7% and 8.7% in terms of Acc@1 and Acc@5, respectively. Similar improvements are also found in methods using ViT backbone.

## 4.4 ABLATION STUDY AND DISCUSSION

**The effectiveness of RDVC.** We analyze the effectiveness of RDVC for both the fine-grained image retrieval and FG-ZS-SBIR tasks. For the former task, we analyze the effect of the RDVC on the

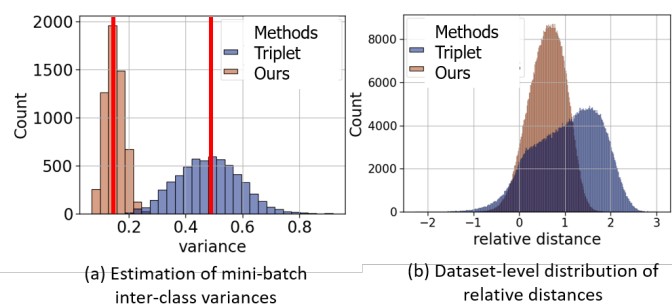

Figure 4: Illustrations of the relevant learned distributions based on testing set of CUB200-2011: (a) mini-batch variance and (b) dataset-level relative distances.

mini-batch interclass variance and the dataset-level distribution of relative distances in Fig. 4(a) and (d), respectively. In Section 3.2, we explain that reducing the relative-distance distribution variance leads to a more uniformly distributed feature space. It can be seen in Fig. 4(a) that the triplet loss with RDVC results in with smaller variations in each mini-batch and a less dispersed distribution of relative distances across the entire dataset, as illustrated in Fig. 4(b). For the FG-ZS-SBIR task, Table 4 gives the ablation study results based on the Sketchy dataset. It can be seen that PK Sampling produce a very similar performance to Random Sampling for baseline (BL) with triplet loss. In contrast to this, our approach significantly improves performance, producing relative improvements from 21.32% to 26.3%, 46.69% to 53.35%, and 59.63% to 66.45% for Acc@1, Acc@5, and Acc@10, respectively, when compared to the baseline (BL) with PK Sampling. Comparative results for different categories are also shown in Fig. 5. It can be seen that RDVC has produced performance improvements for nearly all the categories, and these improvements are relatively uniform across all the categories without any particular bias towards learning a specific category. This indicates that RDVC helps the model learn more generalized and robust features that are beneficial across various categories. Additionally, Table 5 shows that, statistically, the difference between 'Random Sampling + BL' and 'PK Sampling + BL' is not significant ($p > 0.05$), whereas our method significantly outperforms both baseline methods ($p \ll 0.05$).

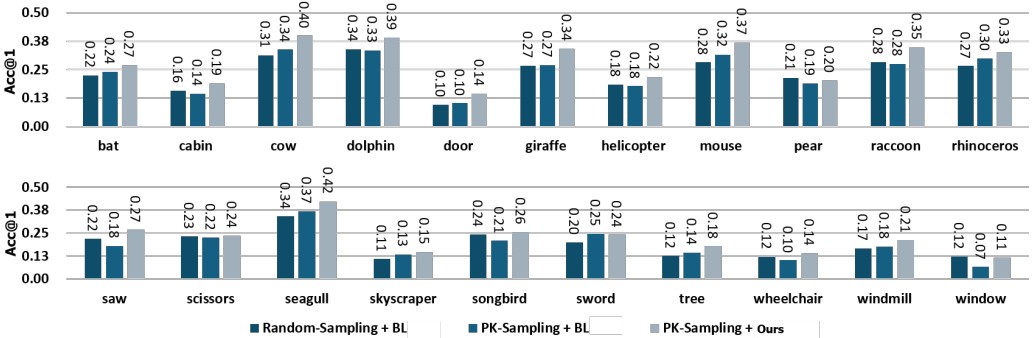

Figure 5: Per-class accuracy of different sampling strategies for the FG-ZS-SBIR task based on the Sketchy dataset. The statistically significant results are presented in Table 5.

Table 4: Effect of sampling strategies for the FG-ZS-SBIR task using the Sketchy dataset.

| Method | Metrics | | |
|---|---|---|---|
| | Acc@1 | Acc@5 | Acc@10 |
| Random Sampling + BL | 21.85 | 47.27 | 60.49 |
| PK Sampling + BL | 21.32 | 46.69 | 59.63 |
| PK Sampling + Ours | 26.30 | 53.35 | 66.45 |

Table 5: A comparison of the statistically significance of the per-class accuracy of the different methods (Fig. 5). A p-value $\ll 0.05$ indicates highly significant difference.

| Paired T-Test | P-value |
|---|---|
| 'Random Sampling + BL' vs 'PK Sampling + BL' | 0.688 |
| 'Random Sampling + BL' vs 'PK Sampling + Ours' | 5.8e-7 |
| 'PK Sampling + BL' vs 'PK Sampling + Ours' | 3.6e-8 |

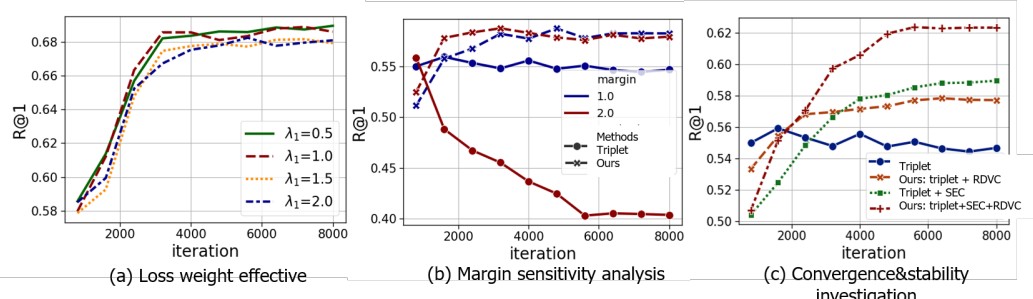

Figure 6: Illustration of various analyses: (a) the effect of loss weight $lambda_1$, (b) sensitivity of margin selection, and (c) convergence and stability.

**The effect of loss weight.** The loss weight $\lambda_1$ in Eq. (10) controls the regularization strength of the RDVC. We analyze in Fig. 6(a) the impact of the loss weight $\lambda_1$ on the performance of FG-ZS-SBIR when assuming different values of $\lambda_1$. As can be seen from Fig. 6(a), our RDVC leads to a robust improvement with $\lambda_1 \in [0.5, 2.0]$. This demonstrates that our RDVC is not sensitive to the choice of $\lambda_1$, with the minimization of the relative distance variance improving the generalization ability.

**Sensitivity analysis of the margin selection.** We analyze the impact of the RDVC on the sensitivity of margin selection using a baseline model with triplet loss on the Cars196 dataset. Fig. 6(b) shows how the R@1 accuracy changes over iterations with and without RDVC under different margin settings. As can be seen, the triplet loss exhibits a high sensitivity to changes in the selected margin. When the margin is increased from 1 to 2, the network converges rapidly within 800 iterations, but subsequently collapses due to an extremely biased learning on training samples. However, the integration of our RDVC loss with triplet loss rectifies the biased learning by regularizing the relative distance and stabilizing the model training. This demonstrates that our RDVC can effectively reduce the reliance on selected margin values in model learning, thereby addressing the problem of having to carefully select the fixed margin.

**Analysis of convergence.** We analyze in Fig. 6(c) the convergence of metric learning with and without RDVC. As can be seen, the model without RDVC converges faster and achieves higher accuracy on the 1000th iteration than the model with SEC and RDVC, and the model with RDVC achieves the best performance at the 2000th iteration. This indicates that the original triplet loss exhibited early-stage overfitting, causing it to overly adapt to those samples with significant differences while neglecting the overall characteristics of the data. By imposing constraints on the feature norm and relative distance by SEC and our RDVC, respectively, this reduces the overfitting problem. Although SEC slows down the convergence speed, the network's learning process accelerates after integrating SEC with the $\mathcal{L}^{RDVC}$, leading to improved performance.

## 5 CONCLUSIONS

In this paper, we develop a novel Relative Distance Variance Constraint (RDVC) loss to regularize pair-distance based deep metric learning (DML). We provide both empirical and theoretical analyses to demonstrate the effectiveness of our RDVC loss. Extensive experimental results on three datasets show that the RDVC loss can ensure a robust and effective representation learning, when combining with other existing loss functions, and it reduces reliance on careful margin selection and increases the chance of selecting informative pairs in the sampling process for model training. Moreover, the RDVC loss has proven effective in FG-ZS-SBIR, a challenging task that requires bridging the gap between different domains and between seen and unseen data. In the future, we will focus on extending the RDVC loss to general cross-domain learning and zero-shot learning.

ACKNOWLEDGEMENTS

The work described in this paper is supported, in part, by a grant from xxx (Grant Number xxx) and by xxx under grant xxx.

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

## A  APPENDIX

### A.1  ILLUSTRATION OF OUR OBJECTIVE FUNCTION

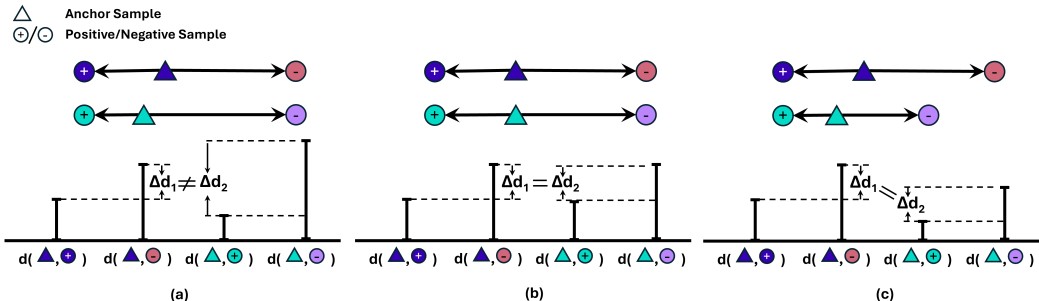

Figure 7: Illustration of our objective function, with the different colors indicating different categories. In (a), it is assumed that there are two hard triplets in a mini-batch, where both triplets are in the condition of $d_{an} - d_{ap} < m$, while the blue-anchor triplet is harder (i.e., the positive distance $d_{ap}$ is closer to the negative distance $d_{an}$) than the green-anchor one. Although both triplets have different levels of hardness (i.e., $\Delta d_1 \neq \Delta d_1$), the triplet loss will generate equivalent gradients with respective to them which is considered unfair. In (b), all distances are forced to be equal to each other for positive and negative pairs respectively, which can alleviate the problem associated with (a). This strategy, however, leads to the loss of class-specific information, disabling any anchor to learn discriminative features. (c) addresses the problems present in both (a) and (b) by incorporating relative distance variance minimization.

### A.2  FULL VERSION OF GRADIENT DERIVATION IN RELATION TO TRIPLET LOSS

Considering $\mathcal{L}^{total} = \mathcal{L}^{tri} + \mathcal{L}^{RDVC}$ , we have

$$\frac{\partial \mathcal{L}^{total}}{\partial \mathcal{D}_i} = \lambda_1 \frac{\partial \mathcal{L}^{tri}}{\partial \mathcal{D}_i} + \lambda_2 \frac{\partial \mathcal{L}^{RDVC}}{\partial \mathcal{D}_i}. \tag{11}$$

Consider the first term, $\mathcal{L}^{tri}$ on the R.H.S of (11)

$$\frac{\partial \mathcal{L}^{tri}}{\partial \mathcal{D}_i} = \frac{\partial}{\partial \mathcal{D}_i} \max(\mathcal{D}_i + \alpha, 0) = \begin{cases} 1, & \text{if } \mathcal{D}_i \geq -\alpha \\ 0, & \text{otherwise} \end{cases} \tag{12}$$

Let's move on to the second term, $\mathcal{L}^{tri}$, of the R.H.S of (11). Since

$$\frac{\partial \mathcal{D}_j}{\partial \mathcal{D}_i} = 0 \text{ for } i \neq j, \frac{\partial \mathcal{D}_j}{\partial \mathcal{D}_i} = 1 \text{ for } i = j, \text{ and } \frac{\partial \mu_{\mathcal{D}}}{\partial \mathcal{D}_i} = \frac{1}{N},$$

We can deduce that

$$\begin{aligned} \frac{\partial \mathcal{L}^{RDVC}}{\partial \mathcal{D}_i} &= \frac{\partial}{\partial \mathcal{D}_i} \frac{1}{N-1} \sum_{j=1}^{N} (\mathcal{D}_j - \mu_{\mathcal{D}})^2 = \frac{1}{N-1} \sum_j 2(\mathcal{D}_j - \mu_{\mathcal{D}}) \frac{\partial(\mathcal{D}_j - \mu_{\mathcal{D}})}{\partial \mathcal{D}_i} \\ &= \frac{1}{N-1} \sum_j 2(\mathcal{D}_i - \mu_{\mathcal{D}}) \frac{\partial \mathcal{D}_j}{\partial \mathcal{D}_i} - \frac{1}{N-1} \sum_j 2(\mathcal{D}_i - \mu_{\mathcal{D}}) \frac{\partial \mu_{\mathcal{D}}}{\partial \mathcal{D}_i} \\ &= \frac{2}{N-1} (\mathcal{D}_i - \mu_{\mathcal{D}}) \end{aligned} \tag{13}$$

Then we have:

$$\frac{\partial \mathcal{L}^{total}}{\partial \mathcal{D}_i} = \begin{cases} \frac{N-1+2(\mathcal{D}_i-\mu_{\mathcal{D}})}{N-1}, & \text{if } \mathcal{D}_i \geq -\alpha \\ \frac{2(\mathcal{D}_i-\mu_{\mathcal{D}})}{N-1}, & \text{otherwise} \end{cases} \tag{14}$$

Hence, every triplet will contribute gradients to enhance the model's learning process.

### A.3 Full Version of Gradient Derivation in Relation to N-Pair Loss

Recall the definition of N-pair loss, we have:

$$\mathcal{L}_{NP} = -\log \left( \frac{e^{1-d_p^i}}{e^{1-d_p^i} + \sum_{j=1}^{L} e^{1-d_n^j}} \right) = \log \left( \frac{e^{1-d_p^i} + \sum_{j=1}^{L} e^{1-d_n^j}}{e^{1-d_p^i}} \right)$$

$$= \log \left( 1 + \sum_{j=1}^{L} e^{d_p^i - d_n^j} \right) = \log \left( 1 + \sum_{j=1}^{L} e^{\mathcal{D}_i} \right) \tag{15}$$

Then we can calculate the gradient w.r.t $\mathcal{D}_i$:

$$\frac{\partial \mathcal{L}^{NP}}{\partial \mathcal{D}_i} = \frac{\partial \log \left( 1 + \sum_{j=1}^{L} e^{\mathcal{D}_i} \right)}{\partial \mathcal{D}_i} = \frac{e^{\mathcal{D}_i}}{1 + \sum_{j=1}^{L} e^{\mathcal{D}_i}} \tag{16}$$

We can see that $\dfrac{\partial \mathcal{L}^{NP}}{\partial \mathcal{D}_i}$ solely focuses on the sum of $L$ relative distances, overlooking the variance within a mini batch, resulting in a similar issue of a distorted decision boundary to the triplet loss.

### A.4 Full Version of Gradient Derivation in Relation to MS Loss

Recall the definition of MS loss in Eq. 4, we can consider a relaxed form by letting $\alpha = \beta = 1$, we then will have:

$$\mathcal{L}_{MS} = \frac{1}{\alpha} \log \left( 1 + \sum_{i=1}^{K} e^{-\alpha(1-d_p^i - m)} \right) + \frac{1}{\beta} \log \left( 1 + \sum_{j=1}^{L} e^{\beta(1-d_n^j - m)} \right)$$

$$= \log \left( 1 + \sum_{i=1}^{K} e^{-(1-d_p^i - m)} \right) + \log \left( 1 + \sum_{j=1}^{L} e^{(1-d_n^j - m)} \right)$$

$$= \log \left( \sum_{i=1}^{K} e^{-(1-d_p^i - m)} \right) + \log \left( \sum_{j=1}^{L} e^{(1-d_n^j - m)} \right) \quad \text{if softplus is disabled} \tag{17}$$

$$= \log \left( \sum_{i=1}^{K} \sum_{j=1}^{L} e^{d_p^i - d_n^j} \right)$$

$$= \log \left( \sum_{i=1}^{K} \sum_{j=1}^{L} e^{\mathcal{D}_i} \right)$$

We observe that Eq. 17 only considers the sum of relative distances (similar to N-pair loss), and does not incorporate variance statistics as constraints.

### A.5 More Experimental Results

Figure 8 presents qualitative results in the FG-ZS-SBIR task, demonstrating the effectiveness of our proposed method.

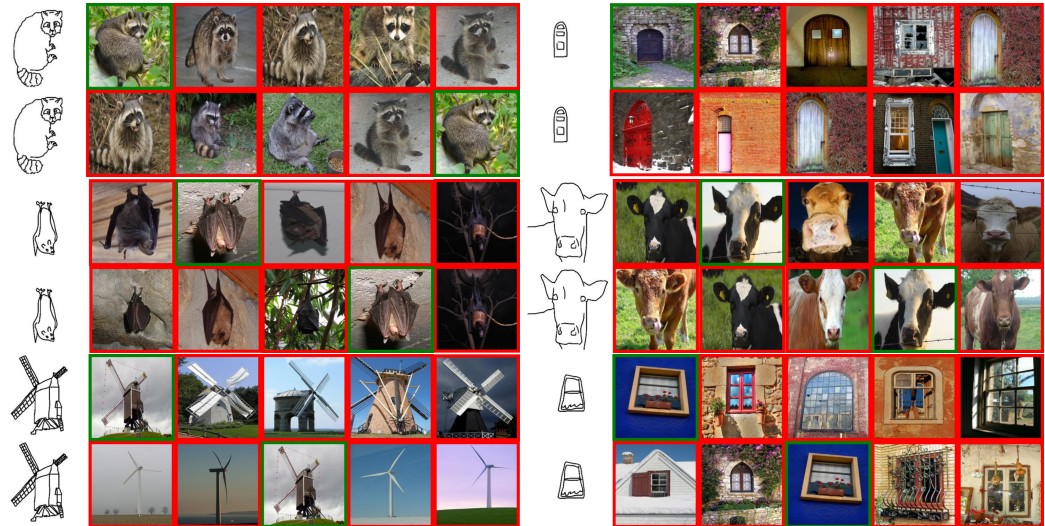

Figure 8: Qualitative comparison. For each query sketch, we show ours (the 1st row of each case) and the baseline (the 2nd row of each case).

## A.6 More Implementation Details

For the the combination of our RDVC and SEC Zhang et al. (2020) loss, we directly employ the default settings from Zhang et al. (2020). Our full settings are shown in Table A.6. As shown, T, SHT, NNP and MS represent triplet Hoffer & Ailon (2015), semi-hard triplet Schroff et al. (2015b), N-pair Sohn (2016) and multi-similarity loss, respectively. $\eta$ is the loss weight for $\mathcal{L}^{SEC}$ and $\lambda_1$ is our loss weight.

| | | | Settings from Zhang et al. (2020) | | Our setting | |
|---|---|---|---|---|---|---|
| Dataset | Iters | Loss | LR Settings (lr for head/lr for backbone/lr decay@iter) | $\eta$ | $\lambda_1$ w/o SEC | $\lambda_1$ w/ SEC |
| CUB200-2011 | 8k | T, SHT
NNP
MS | 0.5e-5/2.5e-6/0.1@5k
1e-5/5e-6/0.1@5k
5e-5/2.5e-5/0.1@3k, 6k | 1.0, 0.5
1.0
0.5 | 2.0, 4e-5
2e-4
0.5 | 0.5, 4e-5
2e-4
1.0 |
| Cars196 | 8k | T, SHT
NNP
MS | 1.5e-5/1e-5/0.5@4k,6k
1e-5/1e-5/0.5@4k,6k
4e-5/4e-5/0.1@2k | 0.5, 0.5
1.0
1.0 | 2.0, 4e-5
2e-4
0.5 | 0.25, 1e-4
5e-5
1.0 |

Table 6: Full version of hyperparameter settings.

