# OpenReview forum: "EFFECTIVE REGULARIZATION WITH RELATIVE-DISTANCE VARIANCES IN DEEP METRIC LEARNING"
_ICLR.cc/2025/Conference — ICLR 2025 Conference Withdrawn Submission_

### Official Review · Reviewer_bNKo · 2024-10-30

**Soundness:** 3
**Presentation:** 2
**Contribution:** 2
**Rating:** 3
**Confidence:** 5

**Summary:**

This paper introduces a regularization method in deep metric learning (DML) using relative-distance variance. The authors argue that traditional pair-distance-based metrics in DML have limitations, such as reliance on the selection of an appropriate margin and the need for informative pair selection, which can lead to poor generalization across tasks. To address these issues, the paper proposes a Relative Distance Variance Constraint (RDVC) loss function that regularizes the variance of relative distances across all pairs in the dataset.

**Strengths:**

The paper shows that the RDVC loss can integrate with various existing basic loss functions, showing good performance in Table 1. The paper is well-structured, with a logical flow from the introduction to related work, methodological development, experiments, and conclusions. The concepts of the defined loss functions are explained clearly, making the paper accessible to readers familiar with DML.

**Weaknesses:**

1. The paper primarily focuses on fine-grained image retrieval datasets like CUB200-2011 and Cars196, and the FG-ZS-SBIR dataset Sketchy. Expanding the experimental validation to include more diverse datasets, such as those with different modalities or domains, could strengthen the claims of general applicability.

2. While the paper includes some ablation studies, more in-depth analysis on the impact of different components of the RDVC loss (e.g., the effect of varying the lambda weights or comparing against other regularization techniques) should be provided for further insights into the model's behavior.

 3. The paper should provide more details on the computational overhead introduced by the RDVC loss, especially when compared to standard DML losses. This information is crucial for applications where computational resources are limited.

4. The paper could benefit from a more comprehensive comparison with the state-of-the-art methods. The most of baselines in Table 2 are out of time.

5. The research approach of this paper is somewhat conventional, and it offers a limited degree of innovation within the field of deep metric learning; the theoretical depth is not particularly profound.

**Questions:**

Please refer to section Weaknesses.

---

### Official Review · Reviewer_5V2U · 2024-11-02

**Soundness:** 2
**Presentation:** 2
**Contribution:** 2
**Rating:** 3
**Confidence:** 4

**Summary:**

This paper presents a method for regularizing deep metric learning (DML) using relative-distance variance. The Relative Distance Variance Constraint (RDVC) loss is formulated based on the variance of the relative distances of each triplet. The authors provide both empirical and theoretical analyses, and experiments conducted on benchmark datasets demonstrate the effectiveness of the proposed method.

**Strengths:**

1. The Relative Distance Variance Constraint (RDVC) loss is formulated based on the variance of the relative distances within each triplet.

2. The writing is clear, and the analysis is thorough.

**Weaknesses:**

It is worth noting that distance variance regularization has been addressed in previous work. For example, Kan et al. proposed a metric variance constraint (MVC) in [R1] to enhance DML performance. The primary distinction between RDVC and MVC is that MVC directly computes the variance of pairwise distances, while RDVC calculates the variance of the relative distances of each triplet. The authors should analyze and compare these two methods more thoroughly, as I believe there may not be significant differences between them. Furthermore, experiments should also be conducted on the SOP and In-Shop datasets, utilizing different backbone networks and loss functions. The methods compared on the CUB and Cars-196 datasets do not represent state-of-the-art approaches.

[R1] S Kan, Z He, Y Cen, Y Li, V Mladenovic, Z He. Contrastive bayesian analysis for deep metric learning, IEEE Transactions on Pattern Analysis and Machine Intelligence 45 (6), 7220-7238.

**Questions:**

See the weaknesses.

---

### Official Review · Reviewer_8YAz · 2024-11-03

**Soundness:** 3
**Presentation:** 3
**Contribution:** 2
**Rating:** 5
**Confidence:** 4

**Summary:**

This work introduces the usage of relative-distance variance to regularize deep metric learning (DML) from both empirical and theoretical perspectives. It presents a straightforward loss function that can be seamlessly integrated with various DML loss functions to enhance feature learning. The experiments conducted are comprehensive, demonstrating that the combination of the proposed loss function significantly improves performance across multiple tasks.

**Strengths:**

1. This paper introduces a novel loss function regularizing the relative-distance variance for deep metric learning. The combination of the proposed loss function with some existing methods could bring obvious performance improvement.

2. The paper provides a comprehensive analysis from both empirical and theoretical perspectives.

**Weaknesses:**

1. It is unclear how to address the issue of selecting informative pairs when using the proposed loss function.
2. The comparison methods used are kind of outdated, and it would be beneficial to include the latest competitors from 2023 and 2024 in Tables 2 and 3 for a more comprehensive comparison.
3. The hyper-parameter λ1 was only evaluated at specific values (0.5, 1, 1.5, 2), but to fully assess its robustness, it is recommended to include a wider range of values (0.001, 0.01, 0.1, 1, 10, 100) across different datasets.

**Questions:**

1. I suggest incorporating some of the latest competitors for a more comprehensive comparison.
2. The proposed regularization performs as a loss function, which could be combined with the existing baselines. However, this work only combines some basic methods. I wonder if the loss function still improves performances when combined with the latest methods.
3. Since the objective of this work is feature extraction, it would be pertinent to include competitors such as CLIP and its fine-tuned variations in the comparison.

---

### Official Review · Reviewer_rmEW · 2024-11-04

**Soundness:** 1
**Presentation:** 3
**Contribution:** 2
**Rating:** 3
**Confidence:** 5

**Summary:**

The paper proposes a novel Relative Distance Variance Constraint (RDVC) regularization loss. The paper observes that despite pair/triplet based objectives optimizing distances for a fixed margin i.e.  distance between nearby dissimilar points should be greater than distance between similar points by a fixed amount,  actual difference in distances between dissimilar (different class) and similar (same class) samples (called relative distance) follows a  distribution with significant variance due to some examples being more challenging than others.
They conduct an empirical study to look at distribution of relative distances in metrics learned by different losses on the CUB200 dataset and observe that metrics with a lower variance in the distribution of relative distances loosely have better NMI/F1 performance. Based on these, they propose a regularization loss that directly minimizes variance of such relative distances among triplets within a batch.

The authors show that their regularization loss (RDVC) improves metric learning performance on CUB200 and Cars-196 when it is used with   several pair based losses, including the MS loss.They also evaluate their method on zero-shot sketch based retrieval where it shows strong performance compared to other non metric learning baselines.

**Strengths:**

**Novelty of Approach:** Analyzing the distribution of relative distances of a dataaset is an interesting and new direction. This could be valuable for tasks requiring homogenous embedding distributions.

**Validation on sketch-based retrieval:** The paper tests the method on zero-shot sketch-based retrieval, which is an interesting aplication of metric learning

**Easy Integration with existing techniques:** RDVC’s compatibility with established losses like triplet, multi-similarity, and semi-hard triplet losses enhances its practicality for broad usage in DML models without significant architectural adjustments.

**Weaknesses:**

**W1 Missing Results for Common Benchmarks:** Evaluation on only small benchmarks like CUB200 and Cars-196 is shown. Very widely used and more challenging benchmarks such as SOP and InShop datasets are not included.

**W2 Lack of motivation for the loss**:

 1.The empirical motivation for the loss is weak (Sec  3.1 trend does not seem monotonic).

2. There is no theoretical motivation given for the loss.
   - In fact, enforcing exactly the same distance between two dissimilar examples irrespective of the underlying class they belong to seems like a poor objective for metric learning. This approach may lead to worse embeddings for datasets with a wide variety of classes that differ from each other by varying degrees (the SOP dataset being one such example).
   - As stated in L288-290, 'easy' triplets have the anchor and positive sample pushed apart from each other. There is no motivation or justification provided for this design, and the empirical results in Sec 3.1 do not convince me that this provides useful supervision to the network.

**W3 Weak Experimental Results:**
The paper claims that the method achieves nearly highest accuracy on the given benchmarks, but misses several newer singificantly better performing baselines.

1. The main results in Table 1 only show that RDVC improves several older metric learning techniques (2019 and older).
2. Image retrieval results Table 2 lacks comparisons with several newer, post 2022 SOTA baselines (see i - iv ), whch display significantly better performance as compared to the proposed method.
3. Table 3 lacks a comparison with any other recent metric learning approach. Comparions with zero-shot CLIP based approaches does not seem fair to me due to a very different training/ model setting used by them.

**W4 Limited ablation study:** One of the main claims of the paper is doing away with the need to select a margin hyperparameter. However, the ablation study to back this claim Sec 4.4 (Fig 6 (b)) only shows results for 2 margin values, making it unconvincing

Given the limited experimental results on standard benchmarks coupled with missing comparisons and a lack of clear motivation for the proposed regularization, I am inclined to reject the paper.

[i] Kim, Sungyeon, Boseung Jeong, and Suha Kwak. "Hier: Metric learning beyond class labels via hierarchical regularization." In Proceedings of the IEEE/CVF Conference on Computer Vision and Pattern Recognition, pp. 19903-19912. 2023.

[ii] Roth, Karsten, Oriol Vinyals, and Zeynep Akata. "Non-isotropy regularization for proxy-based deep metric learning." In Proceedings of the IEEE/CVF Conference on Computer Vision and Pattern Recognition, pp. 7420-7430. 2022.

[iii] Yang, Lu, Peng Wang, and Yanning Zhang. "Stop-gradient softmax loss for deep metric learning." In Proceedings of the AAAI Conference on Artificial Intelligence, vol. 37, no. 3, pp. 3164-3172. 2023.

[iv] Liao, Christopher, Theodoros Tsiligkaridis, and Brian Kulis. "Supervised metric learning to rank for retrieval via contextual similarity optimization." In International Conference on Machine Learning, pp. 20906-20938. PMLR, 2023.

**Questions:**

**Triplet sampling:** For calculating the regularization loss $\mathcal{L}^{RDVC}$, are all triplets taken into consideration ? Or is there any sample mining involved ?


**Normalization of embeddings**: Are the embeddings $\ell_2$ normalized ? Whats the image /crop size used for CUB200 and Cars-196 ?

**Fig 2 (c)**: It would be better to plot the variance vs performance (any of NMI or F1) as the axes to make clear the trend, which does not seem monotonic and is of crucial importance to the paper.

**t-SNE Visualization:** Would the authors consider providing a t-SNE visualization of the learned embeddings across training stages to better illustrate RDVC’s effects on distribution uniformity and separation?

Also see weaknesses above

---

### Note · Authors · 2024-11-27

**Comment:**

Thanks for the comments and we will revise and resubmit with more experimental comparisons next time.

**Withdrawal Confirmation:**

I have read and agree with the venue's withdrawal policy on behalf of myself and my co-authors.